# The role of *Kaizen* in the effectiveness of Amhara metal industry and machine technology development enterprise in Ethiopia, as perceived by the employees

**Melaku Mengistu Gebremeskel**\*

Bahir Dar University, Bahir Dar, Ethiopia

\* [mmelaku25@gmail.com](mailto:mmelaku25@gmail.com)

## Abstract

This study investigated the role of Kaizen in the effectiveness of the Amhara Metal Industry and Machine Technology Development Enterprise in Ethiopia, as perceived by its employees. To this end, the study examined employees' perceptions of the practice and utility of Kaizen, the relationship between Kaizen and organizational effectiveness, and the role of job performance and affective commitment in mediating the effect of Kaizen on the enterprise's effectiveness. The study employed an embedded mixed methods design, with documentary data playing a supportive role alongside the survey data. It involved 247 participants, who were recruited using a systematic random sampling technique to complete a questionnaire. The data were analyzed using both descriptive statistics (mean and standard deviation) and inferential statistics (one-sample t-test, correlation coefficient, confirmatory factor analysis, and structural equation modeling). According to employees' perceptions, although the practice of Kaizen within the enterprise was found to be poor, Kaizen showed a significant relationship with both employee affective commitment and job performance, as well as with the effectiveness of the enterprise. Moreover, Kaizen accounted for approximately 44% of the enterprise's effectiveness, 37% of employee job performance, and 20% of employees' affective commitment. Given that the poor implementation of Kaizen still explained such a significant proportion of the enterprise's effectiveness, it is plausible to conclude that its proper application could make a substantial contribution to productivity and profitability.

## 1. Introduction

Needless to say, organizations today are experiencing unprecedented globalization and dynamic market environments more than ever before [1,2]. As a result, local events are shaped not only by local contexts but also by occurrences that take place many miles away [3,4,5]. According to various sources [2,6], companies that increase

**Data availability statement:** All relevant data are within the manuscript.

**Funding:** The author(s) received no specific funding for this work.

**Competing interests:** The author has declared that no competing interests exist.

their productivity and competitiveness are those that adopt innovative and systematic approaches to production, efficiency, and profitability, based on the ever-changing global context.

The Kaizen technique is one mechanism that should be considered to ensure the competitiveness of organizations in the current market context [7,8,9]. Hosono et al. [10] characterized Kaizen as an inclusive and participatory approach toward continuous improvement of quality and productivity, without requiring additional investment in machinery. It is a production philosophy derived from two Japanese words: *kai* (change) and *zen* (good), meaning change for the better [11,12]. The concept is understood as making small, continuous improvements across various aspects: quality, technology, processes, organizational culture, productivity, safety, and leadership [13,14]. According to these sources, Kaizen makes jobs easier and safer by breaking down tasks and making improvements.

Dahlgaard et al. [15] and Martin and Osterling [12] argue that Kaizen is a process-focused technique, just as important as the intended results. Barnes [16] and Womack and Jones [17] similarly assert that improvements through the Kaizen approach require process-oriented thinking and emphasize people's efforts. That is, Kaizen emphasizes the "how" of meeting required targets – because, in order to achieve improved results, processes must be improved [18].

Thus, Kaizen is a people-oriented practice that fosters continuous, process-oriented, and innovative thinking. According to Goetsch and Davis [13] and Shukla and Ganvir [18], this method posits that the process itself is the target, and employees can drive improvements by understanding how their jobs fit into the process and continuously making changes. For Kaizen to be effectively utilized, managers must focus on employee discipline, involvement, morale, communication, time management, skill development, and other factors. This implies that Kaizen is a management tool aimed at improving productivity, effectiveness, safety, and waste reduction [13]. According to Goetsch and Davis, Kaizen engages and empowers employees at all levels, motivates their morale, uncovers hidden talents, promotes self-development, fosters ownership and trust, reduces waste of time and other resources, and enhances communication and harmony across management levels. In this regard, Kaizen plays a significant role in improving the overall work environment, fostering team spirit, enhancing productivity and profitability, and improving customer satisfaction.

Cheser [19] and Imai [9,20,21] characterize Kaizen by four major features: (1) involving all employees, (2) improving methods or processes of work, (3) focusing on small and incremental improvements, and (4) using teams to achieve incremental changes. Expanding on this idea, Imai [9] asserts that Kaizen is a broad concept that promotes quality and continual improvement at all levels, at all times, forever. A range of literature [8,22] also claims that, as long as Kaizen is in place, all aspects of an organization – people, processes, management, work environment, methods, and products – improve continuously, as there is no concept of "good enough" in Kaizen. Womack and Jones [17] state that Kaizen is a philosophy of never being satisfied with what has already been accomplished. According to them, the job of improvement is never finished, and the status quo is always challenged. The more Kaizen is

practiced, the more it brings together all the employees of a company [11]. Its effective implementation improves communication and strengthens feelings of membership and team spirit, while ensuring that companies can secure customer satisfaction and loyalty [22,23].

Because it minimizes implementation costs, the practice of Kaizen is widely recognized as one of the best methods for performance improvement in companies [23]. It is a valuable tool for increasing productivity, obtaining a competitive advantage, and improving overall business performance in today's tough competitive market environment [23]. Due to its contributions to creating a conducive work environment and ensuring employee safety, Kaizen plays a pivotal role in the productivity and efficiency of companies [13,23].

Principally, Kaizen is represented by five fundamental components known as the 5Ss [13,24]. They represent the first letters of five Japanese words: *seiri* (sort or organize), *seiton* (set in order), *seiso* (shine or scrub), *seiketsu* (systematize or standardize), and *shitsuke* (sustain or self-discipline). Sorting refers to going through all the tools, materials, etc., in the workplace and keeping only the essential items. It involves separating the necessary from the unnecessary and eliminating the unnecessary in areas such as tools, work-in-progress, machinery, products, and documents. *Set in order* involves putting tools and materials in their proper places and keeping things in order so that employees can always find what they need to do the job without wasting time. *Shine* includes three primary activities: cleaning the workplace, maintaining its appearance, and using preventative measures to keep it clean, so that work can proceed efficiently without the disruptions caused by a messy environment. *Standardize* is the phase during which the team identifies ways to establish the improved practices as standard. *Sustain* is a step to maintain the momentum generated during the initial event or project. It involves adhering to standardized work procedures and requires discipline. The aim of the 5Ss is to ensure that only what is necessary is present in the workplace, everything has a designated place, there is a standard way of doing things, and there is discipline to maintain it [25]. Studies [18,23,25] have shown that the effective application of the 5Ss significantly enhances organizational productivity and profitability by driving employees' affective commitment and performance. Accordingly, the 5Ss are the key dimensions addressed by this study.

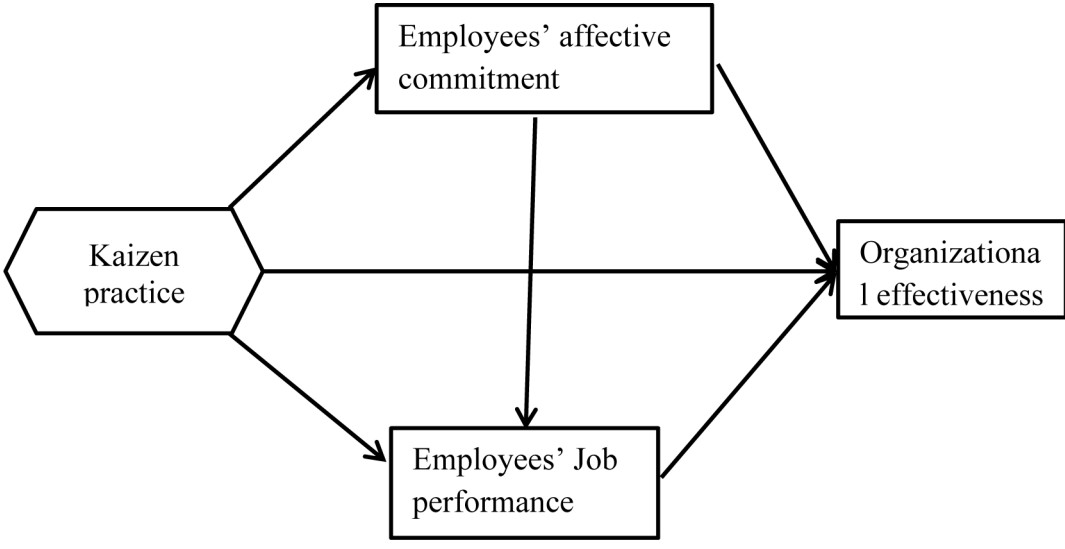

**Fig 1.  *Conceptual Framework of the Study.***

## 2. Problem statement

The Amhara Metal Industry and Machine Technology Development Enterprise (AMIMTDE) is a government-owned enterprise [26]. It was established with the overarching goal of supporting and enhancing the manufacturing industry to realize the structural transformation of the state economy. According to this source, AMIMTDE envisions not only the development and transformation of the quality and competitiveness of enterprises in terms of productivity and utility but also aims to promote and commercialize them on a broad scale. To this end, the enterprise has developed its own Kaizen policy and strategy manual [27], which outlines the employment of Kaizen in every section of the enterprise. This approach aims to continuously improve productivity, customer orientation, and profitability, while also guaranteeing employees better job security and benefits.

By ensuring continuous improvement in the business operations of any enterprise, the Ethiopian Ministry of Industry [28] stipulates that Kaizen ensures organizational effectiveness and customer satisfaction. Accordingly, the government of Ethiopia established the Kaizen Institute "to carry out broad-based activities of ongoing quality and productivity improvement, thereby enhancing the expansion of competitive industries" [[29], p. 6199]. Since its establishment, the institute (named Kaizen Excellence Centre since 2022) has been involved in various activities to introduce and harvest the benefits of Kaizen in both public and non-public organizations across the country. According to its 2017 report, by introducing Kaizen into more than 500 organizations, the institute claims to have saved more than two billion Birr (approximately USD 40 million at the exchange rate at the time) for the country [30].

Despite these claims, empirical and anecdotal evidence suggests that the practice of Kaizen has been insufficient in Ethiopia. In contrast to the Kaizen Institute's report, Asayehgn [31] reported that little had been accomplished beyond posting the 5Ss on the walls of enterprises. In other words, companies did not properly arrange their work areas as required to effectively enhance employee commitment, performance, and organizational productivity. Asayehgn characterized these companies as superficially adopting Kaizen, merely using its terminology to appear competent, but ultimately achieving little – what he referred to as "paper tigers." A year later, Asayehgn et al. [32], in their analysis of Kaizen implementation in manufacturing enterprises in northern Ethiopia, reported that company executives were neither committed to Kaizen teamwork nor did they encourage front-line workers to collaborate toward Kaizen's effectiveness.

Four years later, Abraham and Singh [33], who conducted research on the implementation of Kaizen in manufacturing firms in Ethiopia, claimed that many Ethiopian firms lack awareness of Kaizen principles due to insufficient training opportunities, a knowledge gap he attributed to hampering the effective adoption and integration of Kaizen practices. As a consequence, he also expounded that organizational resistance to change was prevalent, with employees and management often hesitant to adopt new methodologies, which impeded the successful implementation of Kaizen initiatives. In a similar study conducted in the same year on firms in Ethiopia, Abebe [34] asserts that the traditional Ethiopian work culture, identified by Yabibal and Tibletie [35], Alie et al. [36] and Haapatalo [37] as one of the determining factors, does not align seamlessly with Kaizen's emphasis on continuous improvement and employee involvement, leading to implementation difficulties. Fasika and Alemayehu [38] conducted a study on Kaizen implementation in industries in southern Ethiopia, which revealed some contradictory findings but confirmed the challenges of implementing Kaizen effectively. Like Asayehgn et al. [32], the authors found a lack of commitment from both executives and front-line workers, a finding also indirectly supported by Jin [23]. Fasika and Alemayehu [38] also argue that a lack of necessary resources, including financial support and infrastructure, poses a significant challenge to the practice of Kaizen, without which sustaining its practices becomes difficult.

Jin [23], a government-supporting author, on the other hand, reported that "the government has a clear vision that Kaizen needs to be promoted by the public sector" (p. 108). These conflicting arguments between government agencies and research reports raised significant concerns, prompting the researcher to select a specific government enterprise, AMIMTDE, to examine employees' perceptions of the practice and utility of Kaizen.

To this end, AMIMTDE's ten-year strategic plan (2022–2031) [39] and the subsequent operational plans and performance reports were reviewed to assess how Kaizen was incorporated into the enterprise's operations. However, unlike the Kaizen policy and strategy manual, the strategic plan of the enterprise did not provide sufficient attention to Kaizen.

Consistently, the subsequent performance reports of the enterprise presented unclear content regarding the role of Kaizen in improving its effectiveness. For example, despite claiming 94.8% achievement in sorting, the 2021/22 performance report displayed only 73% achievement in sustaining, even though Asayehgn et al. [32] had previously reported that no enterprise in Ethiopia had fully achieved the goals of standardization and sustaining in Kaizen. The same pattern was observed with other components of the 5Ss. Moreover, the report indicated that Kaizen case teams (known as *kelbus* in Kaizen terminology) had held fewer than 42% of the planned discussion sessions, and those that did occur were not problem-solving in nature. The 2022/23 performance report completely disregarded Kaizen.

Additionally, the 2021/22 performance report criticized the lack of close follow-up and continuous identification of problems, as well as the absence of devised solutions by *kelbus* teams. Above all, it disclosed the persistence of significant waste issues within the enterprise. Based on these performance reports, it can be concluded that AMIMTDE did not implement Kaizen properly and sustainably. A 94.8% achievement in sorting and 73% in sustaining is therefore highly questionable under these conditions. This suggests that the enthusiasm of both executives and line employees for Kaizen is open to doubt. The purpose of this study, therefore, is to examine the perceptions of AMIMTDE employees regarding Kaizen and its utility in enhancing organizational effectiveness. To that end, the following research questions guide the study:

1. What do the perceptions of AMIMTDE employees reveal about the practice and utility of Kaizen in enhancing the effectiveness of their organization?

2. To what extent does the practice of Kaizen explain the effectiveness of AMIMTDE?

3. To what extent do job performance and the affective commitment of employees mediate the effect of Kaizen on the effectiveness of AMIMTDE?

## 3. Methodology

### 3.1. Research design

This study employed a sequential embedded mixed-methods design that helps to draw more rigorous conclusions [40,41]. A wide range of literature [42,43] advocate that using a single data source and method may not suffice to arrive at valid conclusions, as compared to using different data sources and employing multiple data collection methods. That is because, according to both these sources, the method allows for richer insights, as the qualitative data can help explain or deepen understanding of the patterns found in the quantitative data. The design effectively explicates and fine-tunes employee perceptions with evidences documented in the enterprise. According to these sources, information obtained from different data sources and methods enables the crosschecking (triangulation) of findings and provides a more comprehensive answer to the research questions. In the context of the current study, incorporating data extracted from documentary examination strengthens the validity of the research, as it not only reduces biases and increases confidence in the results but also provides more robust and well-rounded arguments for the findings. On this basis, data in this study were collected through an objective questionnaire and document analysis, with the latter serving a supportive role to the former. Documentary analysis was conducted after collecting and analyzing the data obtained through the questionnaire.

The target population of the study, which is equivalent to the sampling frame, consisted of all employees of AMIMTDE, totaling 520 individuals [44]. To determine the sample size, Yamane's [45] formula for a 95% confidence level was used. Accordingly, the sample size n was calculated as follows:

$$\frac{N}{1 + N\left(e^2\right)}$$

*Where, N is the population size and e is the level of precision*.

According to this formula, the sample size for the current study was approximately 226. To address challenges such as unreturned questionnaires, incomplete responses (e.g., missing items, multiple ticks in a single row of choices), or participant attrition, the sample size was increased by a 10% non-response insurance factor [40]. This adjustment brought the sample size to around 247, calculated as follows: 226 + (226 x 0.1). These 247 participants were selected using systematic random sampling, based on the employee attendance sheet regularly utilized by the enterprise. The sample size was deemed sufficient, as scholars in the field – such as Innami and Koizumi [46] and Collier [47] – suggest that a sample size 10 times the number of parameters is adequate for running structural equation modeling (SEM). In this study, with 17 parameters (10 x 17), the sample size of 247 exceeds the minimum required.

### 3.2. Instruments

This study aimed to measure employee perceptions of the practice and role of Kaizen in AMIMTDE. To achieve this, a questionnaire was used as one of the data collection methods, as it is commonly required in survey design studies [48]. The questionnaire consisted of both adopted and self-prepared items. The items related to employee commitment were adopted from Meyer and Allen [49], while the items for the other three variables were self-developed based on the literature reviewed. Since the study sought to measure employees' determination to practice Kaizen, it focused on affective commitment (AC), exclusively considering AC as a key variable. The items designed to measure organizational commitment (OC) were therefore based solely on the AC measurement scales. According to Allen and Meyer [50], AC encompasses three key elements that bind employees to the organization: the development of psychological affinity with the organization, their attachment to the organization, and their active involvement in organizational activities. These elements collectively contribute to better performance and organizational effectiveness. Consequently, the items measuring employee commitment in this study specifically reflected the affective commitment dimension of Meyer and Allen's [49] model.

In addition to the questionnaire, data were also collected through documentary examination. The Kaizen policy, the strategic plan in effect, and the performance reports of the enterprise were reviewed. The data gathered from these documents were integrated with the quantitative data during analysis.

With the exception of a few demographic questions, most items in the questionnaire were close-ended. All variables were measured using two types of attitude scales – Likert and rating scales – with scores ranging from 1 to 5. This approach aligns with the recommendations of scholars such as Cohen et al. [51], Creswell [40], Gay et al. [48], and Mertens [41], who suggest that such scales are easier to score and generally less burdensome for participants to complete. Although the sample size was initially set at 247, approximately 300 questionnaires were distributed to meet the requirements of structural equation modeling (SEM), which typically requires a larger sample size [47,52,53]. As a result, 53 additional questionnaires were distributed to minimize the impact of poor response rates and missing data. Out of the 300 distributed questionnaires, 254 (84.7%) were completed and returned, yielding a satisfactory return rate for data analysis. However, 19 (7.7%) of the returned questionnaires were excluded during data cleaning due to missing items, incomplete responses, or multiple ticks in a single row of choices. Therefore, 235 questionnaires (approximately 95% of the target sample size) were deemed usable for analysis. This return rate was sufficient to proceed with the data analysis.

### 3.3. Validity and reliability

This study used a self-prepared questionnaire that included four items focused on the demographic characteristics of participants and 46 items related to the study's main topic. Of the latter, 28 items addressed the 5Ss, seven focused on employee job performance, six on employees' affective commitment, and five on organizational effectiveness (see Table 1). To refine the factors in accordance with the variables they were intended to measure, the researcher employed principal component analysis (PCA). The varimax rotation was applied through the SPSS dimension reduction tool, which

**Table 1.** *Scale reliability and confirmatory factor analysis* (at p<0.001).

| Latent variable | Item | Factor loading scores | Composite reliability |
|---|---|---|---|
| Kaizen practices | Sorting | .805 | .850 |
| | Setting | .830 | .859 |
| | Shining | .775 | .865 |
| | Systematizing | .800 | .913 |
| | Sustaining | .734 | .917 |
| Employee affective commitment | AC6 | .518 | .824 |
| | AC5 | .625 | |
| | AC4 | .665 | |
| | AC3 | .779 | |
| | AC2 | .739 | |
| | AC1 | .716 | |
| Employee job performance | JP1 | .774 | .940 |
| | JP2 | .830 | |
| | JP3 | .841 | |
| | JP4 | .881 | |
| | JP5 | .854 | |
| | JP6 | .773 | |
| | JP7 | .811 | |
| Organizational effectiveness | OE1 | .703 | .896 |
| | OE2 | .768 | |
| | OE3 | .780 | |
| | OE4 | .844 | |
| | OE5 | .875 | |
| Kaiser-Meyer-Olkin Measure of Sampling Adequacy. | | .858 | |
| Bartlett's Test of Sphericity | Approx. Chi-Square | 1003.343 | |
| | df | 28 | |
| | Sig. | .000 | |
| | Eigenvalue | 1.176 | |

helped maximize the dispersion of factor loadings within the variables. In line with the guidelines provided by Field [54] and Tabachnick and Fidell [55], this method was useful in identifying which items measured each variable independently. It also helped to summarize patterns of correlations among the observed variables, reduce the number of variables into a smaller number of factors, and provide a clearer understanding of the relationships between the observed variables and the factors they were intended to measure.

Additionally, to assess the relevance and trustworthiness of the research, the validity and reliability of the items were tested before the actual data collection took place. To evaluate the validity, Lawshe's [56] content validity model was applied. This model is useful for determining how well each item measures the specified variables. Known as the content validity ratio (CVR), it helped to decide whether to reject or retain specific items. According to Lawshe, the CVR is calculated as follows:

$$\frac{n_e - N_2}{N_2}$$

Where $n_e$ is the number of panelists indicating "essential" and N is the total number of panelists.

To test the scale's reliability and validity, the draft questionnaire was distributed to ten employees who served as panelists to assess the validity of the items. These panelists were excluded from the final study sample. Following Lawshe's [56] guidelines, the items were rated on a two-point scale: 1 = not essential and 2 = essential. Based on their ratings, five items with poor validity were removed, and six others were modified.

In addition to the validity test, the reliability of the scale was assessed for each variable through internal consistency testing. In reporting the results, items related to a specific variable were grouped together, and their composite scores were used instead of reporting individual items. As shown in Table 1, the internal consistency for each variable demonstrated values higher than 0.70, which is the minimum reliability threshold suggested by several scholars [57–59].

Furthermore, an extensive reliability analysis was conducted using confirmatory factor analysis (CFA) and factor loading methods for each factor of the measured variables. As presented in Table 1, the factor loadings for all items ranged from 0.518 (item AC6) to 0.881 (item JP4), all of which meet Hair et al.'s (2010) rule of thumb that an item is considered significant if its factor loading is greater than 0.50.

In addition to the validity and reliability tests, the questionnaire items were further refined through exploratory factor analysis (EFA). A data reduction technique, principal component analysis (PCA), was then employed to simplify the subsequent analysis by replacing a larger number of original variables with a smaller number of composite variables (Kline, 2016). This approach, as recommended by Cohen et al. [51] and Tabachnick and Fidell [55], helps identify patterns and group underlying factors that measure the intended variables. Additionally, as suggested by Field [54] and Tabachnick and Fidell [55], PCA helped retain factors with large eigenvalues while removing those with small eigenvalues, thereby optimizing the scale's reliability.

In this respect, factor analysis was initially performed to assess the suitability of the data for factor analysis. The correlation matrix, along with the Kaiser-Meyer-Olkin (KMO) Measure of Sampling Adequacy and Bartlett's Test of Sphericity, demonstrated satisfactory results. Specifically, the KMO value exceeded 0.60, and Bartlett's test of sphericity yielded statistically significant results, indicating that the data were appropriate for factor analysis.

Regarding normality, the skewness and kurtosis values were examined. As shown in Table 2, all skewness and kurtosis values were within the acceptable limits outlined by Kline [53]. According to his more conservative guidelines, skewness should not exceed 3.0 and kurtosis should not exceed 10.0. The data, therefore, met these criteria and did not violate the assumptions required for conducting structural equation modeling (SEM).

**Table 2. Test of normality (N = 235).**

| Variable | Skewness | | Kurtosis | |
|---|---|---|---|---|
| | Statistic | S. E. | Statistic | S. E. |
| Sorting | .080 | .159 | -.612 | .316 |
| Setting | -.067 | .159 | -.270 | .316 |
| Shine | .140 | .159 | -.064 | .316 |
| Systematize | .388 | .159 | -.020 | .316 |
| Sustain | .411 | .159 | .011 | .316 |
| Job performance | -.171 | .159 | -.882 | .316 |
| Organizational effectiveness | .325 | .159 | -.573 | .316 |
| Affective commitment | -.189 | .159 | -.682 | .316 |

### 3.4. Data analysis method

Once the data were collected, they were coded and entered into IBM SPSS-26 and IBM SPSS AMOS-24 software [60] for analysis. The adequacy of response rates was assessed, and nonconformities, such as incomplete questionnaires, missing items, attrition, and errors (e.g., multiple ticks in a row of choices instead of just one), were addressed. Additionally, the data were screened for missing values and outliers. Tests for linearity, homoscedasticity, and multivariate normality (including skewness and kurtosis) were performed to ensure the data met the requirements specified by Byrne [52] and Kline [53]. Next, the maximum likelihood estimation technique was applied to estimate the factor loadings and test the statistical significance of the correlation coefficients between factors.

Finally, both descriptive and inferential statistics were used, supported by effect size tests. Descriptive statistics included means and standard deviations, while inferential statistics involved one-sample t-tests, correlation coefficients, confirmatory factor analysis (CFA), and structural equation modeling (SEM) [48,53]. CFA and SEM were particularly useful for testing the hypothesized relationships among the variables and assessing the goodness of fit. A satisfactory goodness of fit indicates how well the model aligns with the data and supports the plausibility of the hypothesized relationships among the variables [53].

### 3.5. Ethical symmetry

This study collected data from employees who were all adults. Data were gathered through questionnaires and document examination in November and December 2023 personally by me. The questionnaire included instructions informing participants not to include their names or any information that could disclose their identities or personal details. Before data collection, participants were clearly and precisely informed about the content and purpose of the study. They were also given the right to agree or disagree with completing the questionnaire. Subsequently, they gave the researcher verbal consent regarding their willingness to participate in the study or to answer the items in the questionnaire. On the other hand, the document review focused on the enterprise's performance report, which did not reveal anyone's identity. By doing so, the researcher believes the ethical consent of the study participants was ensured.

## 4. Results and discussion

### 4.1. Practice of *Kaizen* in AMIMTDE

Participants' perceptions of the extent to which both the exogenous (Kaizen) and endogenous (affective commitment, job performance, and organizational effectiveness) variables were practiced were measured using a one-sample t-test statistic. According to Green and Salkind [61] and Martin and Bridgmon [62], this statistic is useful for comparing an obtained mean score with a constant (the test value) to determine if the mean score significantly differs from the constant. The one-sample t-test is a parametric inferential statistic [63] that assesses whether the mean of a test distribution differs significantly from a midpoint, average score, population mean, or a preset value [57,64]. In this study, the midpoint (3) was used as the test value.

The study focused on the composite scores of the latent variables, rather than their individual indicators, to compare mean scores against the test value. The exogenous variable (Kaizen) was decomposed into its five dimensions (5Ss), while the endogenous variables were analyzed as they were. The scores presented in Table 2 indicate that the mean scores were generally below the midpoint (3.00), suggesting poor practice of Kaizen in AMIMTDE. Additionally, the mean score difference was modest for some Kaizen components (setting and shining) but substantial for the remaining three components, further indicating weak implementation of Kaizen. The strong deviations of the mean scores from the average (see effect size test scores in Table 3) imply that Kaizen practice in AMIMTDE is significantly underperforming.

**Table 3. One-sample t-test for the Practice of Kaizen and Organizational Effectiveness (N = 235).**

| Variable | | Test value = 3 | | | | | |
|---|---|---|---|---|---|---|---|
| | | Mean | SD | Mean Difference | t | Sig. (2-tailed) | Effect size |
| Exogenous variable | Sort | 2.5626 | 0.8369 | -0.4374 | -8.0120 | .000 | -0.5226 |
| | Set | 2.6936 | 0.8018 | -0.3064 | -5.8581 | .000 | -0.3821 |
| | Shine | 2.7007 | 0.7995 | -0.2993 | -5.7388 | .000 | -0.3744 |
| | Systematize | 2.4589 | 0.8186 | -0.5411 | -10.1330 | .000 | -0.6610 |
| | Sustain | 2.3312 | 0.8150 | -0.6686 | -12.5760 | .000 | -0.8204 |
| Endogenous Variables | Affective commitment | 3.0093 | 1.2365 | 0.0093 | 0.1153 | .000 | 0.0075 |
| | Job performance | 3.0809 | 0.6943 | 0.0809 | 1.7862 | .000 | 0.1165 |
| | Organizational effectiveness | 2.4485 | 1.0784 | -0.5515 | -7.8397 | .000 | -0.5114 |

This finding aligns with the findings of Abraham and Singh [33], Asayehgn [31], and Fasika and Alemayehu [38], all of who noted that Kaizen was not effectively implemented in Ethiopian enterprises. Basic Kaizen components (such as the 5S system) were often displayed on walls for appearances rather than being truly practiced. Despite the formal introduction of a Kaizen policy and strategy [27] and the establishment of steering committees and sub-committees to facilitate its implementation, employees perceive that little has been accomplished beyond policy development.

This mirrors not only Abraham and Singh's [33] finding that resistance to change impeded the successful implementation of Kaizen initiatives but also Asayehgn's [31] conclusion that Kaizen was largely superficial in its implementation, limited to mere poster displays rather than real organizational change. In contrast to the views of Goetsch and Davis [13] and Shukla and Ganvir [18], the current study suggests that Kaizen has not been fully internalized by the employees of AMIMTDE. The findings also contrast with the recommendations of Imai [9,21] and Cheser [19], who argue that Kaizen requires active involvement of all levels of employees, a team-oriented approach, and strong organizational commitment. In AMIMTDE, however, Kaizen was not effectively supported by functional responsibilities for improvement, and it lacked a true team-based approach and organizational commitment to change.

Regarding the endogenous variables, the findings present a mixed picture. The mean score differences between the employees' perceptions and the test value were relatively weak for employee affective commitment and job performance. However, the differences were significantly stronger with respect to organizational effectiveness (see Table 3).

## 4.2. Role of *Kaizen* on organizational effectiveness

This study aimed to estimate both the direct and indirect effects of Kaizen on the effectiveness of AMIMTDE, with a focus on the mediating roles of job performance and affective commitment brought about by Kaizen. To this end, the first step was to examine the correlation between the exogenous (Kaizen) and endogenous (employee commitment, job performance, and organizational effectiveness) variables (see Table 4). The study employed the 5Ss, as outlined by various scholars [13,22,24], as the measurement scales for Kaizen, as shown in the table.

As demonstrated in Table 4, each Kaizen component is significantly related to employee affective commitment, job performance, and organizational effectiveness (p < 0.01). This substantiates the findings of Abebe [65]. The relationship is also mirrored by the significant association between employee commitment, job performance, and organizational effectiveness. These findings underscore the strong connection between Kaizen, employee commitment, and organizational effectiveness, consistent with findings from a wide body of evaluative research conducted across various enterprises globally [18,21,23,25].

Table 4. Coefficient of correlation between the exogenous and endogenous variables.

| Variables | Employee affective commitment | Employee job performance | Organizational effectiveness |
|---|---|---|---|
| Sort | .299** | .329** | .395** |
| Set | .293** | .378** | .380** |
| Shine | .358** | .371** | .300** |
| Systematize | .378** | .390** | .501** |
| Sustain | .309** | .333** | .484** |
| Employee affective commitment | | .498** | .523** |
| Employee job performance | | | .416** |

**Correlation is significant at the 0.01 level (2-tailed)

The findings displayed in Tables 3 and 4 revealed inconsistent situations in the practice of Kaizen at AMIMTDE. On one hand, employees felt that the overall performance of Kaizen in the enterprise was poor (significantly below average), with mean scores showing moderate to strong deviations from the average. On the other hand, the correlation and regression analyses revealed a strong effect of Kaizen on employee affective commitment and job performance, as well as on organizational effectiveness. These contradictory findings suggest several explanations that align with earlier research. As Asayehgn [31] rightly stated, this confirms that possessing an optimistic perception does not necessarily mean a willingness to embrace new processes or adapt to the Kaizen approach. Optimism alone does not guarantee the effective implementation of Kaizen, as employees may still harbor unconscious resistance to change. This resistance may arise because employees believe their current processes are sufficient, making them reluctant to fully engage with Kaizen initiatives. Alternatively, employees may be overly optimistic but lack the practical knowledge of how to apply Kaizen, especially if there is insufficient communication or understanding of how the process works. Without proper guidance, optimism may not translate into actionable improvements, and employees may fail to see tangible results.

Consistent with the findings of Abraham and Singh [33] and Shukla and Ganvir [18], leadership may not have provided adequate support and resources for Kaizen. Optimistic employees might have high expectations, but without leadership buy-in and consistent encouragement, their enthusiasm may have waned over time, reducing the overall effectiveness of Kaizen initiatives. In line with Haapatalo et al. [37], moreover, the work culture or environment may not have fully supported Kaizen practices, even if employees were optimistic. Alternatively, employees who are overly optimistic may underestimate or overlook problems within the enterprise, making them less likely to engage in the continuous problem-solving process that Kaizen requires. Kaizen is based on identifying and addressing small, incremental problems, but overly optimistic employees might miss critical issues, limiting the potential for improvement. Furthermore, as noted by Shukla and Ganvir [18], Kaizen relies heavily on continuous evaluation and feedback. The absence of a system to monitor progress and hold individuals accountable might have led to ineffectiveness and a lack of tangible improvements. In other words, optimistic perceptions alone did not generate sustained progress because the right metrics and accountability structures were not in place.

Be that as it may, convergent and discriminant validity tests were also conducted to assess the degree of relationship and distinctiveness among the variables, respectively. Convergent validity was measured through statistical techniques such as correlation and regression, in line with Cohen et al. [51]. To assess discriminant validity, two methods were employed. First, following Cohen et al. [51] and Kline [53], a discriminant validity test was conducted by comparing the square root of the average variance extracted (AVE) for each variable, which was found to be greater than the inter-construct correlations. Second, as recommended by Kline [53], the factor loadings of the items or indicators within a construct were examined. The results demonstrated that each item loaded more highly within its intended construct than in any other construct, indicating strong discriminant validity. Both methods of discriminant validity confirmed that the instrument effectively distinguished between the individual constructs.

To assess the role of Kaizen in facilitating organizational effectiveness, both directly and indirectly (through the mediation of employee commitment and job performance), structural equation modeling (SEM) was employed, as illustrated in Fig 1. The first step in this analysis was to examine the model fit indices. According to Collier [47], this step is critical in model evaluation as it helps determine the adequacy of the model in explaining the data. In line with Byrne [52] and Kline [53], the model fit was evaluated using comparative fit indices and residual fit indices, in addition to the commonly employed chi-square test. The comparative fit indices compare the new model with the null model (the model developed beforehand), while the residual fit indices emphasize the average difference between the observed and hypothesized variance/covariance matrices.

Kline [53] and Tabachnick and Fidell [55] justify the use of these two indices for two primary reasons. First, chi-square tests can be less reliable in large sample sizes, as they may flag trivial differences as statistically significant. Second, the comparative and residual fit indices are more appropriate in the context of this study. Therefore, alongside the chi-square test (with its degrees of freedom), the following fit indices were employed: the comparative fit index (CFI), the normed fit index (NFI), the Tucker-Lewis index (TLI), the standardized root mean square residual (SRMR), and the root mean square error of approximation (RMSEA). All parameter estimates presented in the analysis are standardized. As shown in Table 5, the data demonstrated a good fit across all parameters to the hypothesized model (Fig 1), allowing for further analysis.

The effect sizes of one latent variable on another were measured based on the direct and indirect structural relationships, as depicted in Fig 2. The largest direct effect in the model was found to be that of Kaizen on employee affective commitment ($\beta = 0.45$), followed by the effect of employee affective commitment on both job performance ($\beta = 0.41$) and organizational effectiveness ($\beta = 0.41$). Interestingly, the effect of employee job performance on organizational effectiveness ($\beta = 0.12$) was the smallest.

Similarly, the combined direct and indirect effect of Kaizen on job performance ($\beta = 0.48$) suggests that Kaizen has a moderate effect on both employee affective commitment and job performance. While employee affective commitment had a moderate effect on both job performance and organizational effectiveness, employee job performance had only a modest effect on organizational effectiveness. This indicates that, in the context of AMIMTDE, employee affective commitment had a stronger impact on organizational effectiveness than did employee job performance.

Additionally, the direct ($\beta = 0.40$) and indirect ($\beta = 0.05$) effects of employee affective commitment on organizational effectiveness totaled $\beta = 0.45$, signifying a moderate effect. The combined (direct and indirect) effect of Kaizen on organizational effectiveness was notably stronger ($\beta = 0.52$), indicating that, from the employees' perspective, Kaizen – when properly promoted and managed by both managers and employees – has a substantial impact. This finding aligns with Abebe [65], who identified a positive correlation between the practice of Kaizen factors and the performance of manufacturing companies in Ethiopia. The current finding also supports earlier research [13,14,66], which argues that Kaizen makes jobs easier and safer by breaking down tasks and making improvements.

Moreover, the explanatory power of the research model (Fig 1) was assessed by calculating the coefficient of determination ($R^2$) for the endogenous constructs. The results showed $R^2$ values of 0.20, 0.37, and 0.44 for employee affective commitment, employee job performance, and organizational effectiveness, respectively (see Fig 2). This means that Kaizen accounts for 20%, 37%, and 44% of the variance in these variables, according to employees' perceptions. This insight could guide AMIMTDE in refining or expanding its Kaizen implementation to maximize its positive effects across all three areas. This finding aligns with a range of research, including Aamer et al. [66], Ratnawati et al. [25], Jin [23], and Shukla and Ganvir [18], who asserted that the effective application of the 5Ss significantly enhances employees' affective commitment and performance, which, in turn, drive the productivity and profitability of organizations. While Kaizen positively impacts employee commitment, job performance, and organizational effectiveness, it is not the sole determining factor, as other factors such as training, leadership, and resources also play a role in improving employee commitment and job performance.

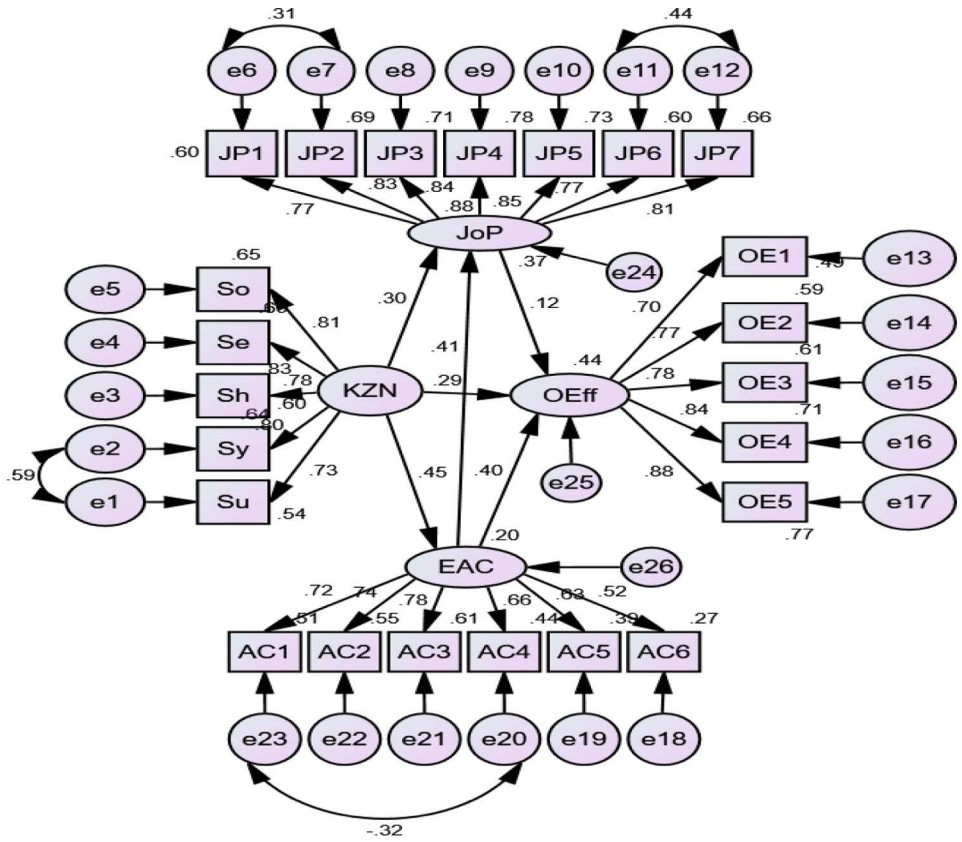

KZN = Kaizen; JoP = Job perfromance; OEff = Organizational effectiveness; EAC= Employee affective commitment

**Fig 2. SEM with Standardized Estimates.**

**Table 5. Summary of goodness-of-Fit Statistics (N = 235).**

| Index | χ² | df | χ²/df | SRMR | RMSEA | CFI | NFI | TLI |
|---|---|---|---|---|---|---|---|---|
| Value | 345.594 | 220 | 1.571 | .046 | .048 | .964 | .916 | .959 |

*Note: χ2 is manipulated at p < 0.001*

Kaizen accounts for a relatively larger portion of changes (44%) in the enterprise's effectiveness, meaning its practices have a greater impact on some areas (such as organizational effectiveness) than on others (like employee commitment). In other words, Kaizen is a key contributor to nearly half of the improvements in the operation of AMIMTDE, as perceived by employees. Employees view Kaizen as a major factor in enhancing organizational effectiveness, indicating that adopting its principles leads to improved processes, better problem-solving, and stronger teamwork – elements that contribute to the overall success and functioning of the enterprise. Consistent with Haapatalo et al.'s [37] findings, a significant portion of AMIMTDE's effectiveness is attributed to the commitment and work culture for continuous improvement practices and principles promoted by Kaizen.

Similarly, explaining 37% of the variance in employee job performance suggests that Kaizen practices likely have a stronger impact on improving job performance, as employees continuously refine their skills and processes, boosting overall performance. Its role in explaining 20% of the variance in employee commitment suggests that Kaizen fosters a sense of ownership and involvement in improving work processes, which could lead to higher levels of commitment from employees. This is consistent with the findings of Patel and Patel [67], who argued that Kaizen promotes employee engagement and empowerment by creating an environment where everyone feels responsible for sustainability.

If Kaizen can explain 44% of organizational effectiveness even in a context where its practice is suboptimal, one can imagine how much more impactful effective Kaizen implementation could be for AMIMTDE's fruitfulness. However, documentary sources [68] reveal that the enterprise achieved only 37.2% of its target business transaction volume (1.55 billion Birr), which was considered very low performance. In its 2023 performance report [44], the enterprise reported achieving only 52.8% of its job performance target. The same report showed that it met just 68.45% of its resource utilization efficiency goals and 13.15% of its profit improvement targets.

While scholars such as Cheser [19], Dahlgaard et al. [15], and Martin and Osterling [12] stress the importance of Kaizen in achieving desired outcomes, the enterprise's recent reports [44,65] largely overlooked its role in these achievements. The reports included only brief and unedited information about Kaizen. Despite the quantitative findings in this study, which indicate that Kaizen explains 44% of organizational effectiveness, the enterprise's lack of a consistent and effective Kaizen practice may have contributed to declines in customer satisfaction, profitability, and overall efficiency.

Earlier studies by Imai [21], Ratnawati et al. [25], Jin [23], and Shukla and Ganvir [18] suggest that organizational productivity and profitability are difficult to achieve without the effective application of the 5Ss and the consequent improvement in employee commitment and performance. However, the reluctance to fully embrace Kaizen at AMIMTDE – both in rhetoric and in practice – has likely hindered its potential. This finding complements Asayehgn's [31] report, which disclosed that the enterprises he investigated in Ethiopia have made only minimal progress beyond simply posting the 5Ss on the walls in their Kaizen practice. As Goetsch and Davis [13] and Jin [23] argue, the insufficient dedication to Kaizen has consistently deprived AMIMTDE of a critical competitive advantage in today's increasingly competitive market environment.

## 5. Conclusion

The data analysis in this study revealed somewhat contradictory findings. On one hand, participants reported poor performance of Kaizen in the enterprise. On the other hand, the correlation and regression analyses indicated that participants held an optimistic view of Kaizen's potential utility, revealing a strong relationship between Kaizen and its outcomes such as employee affective commitment and organizational effectiveness. These mixed results suggest that, while there is a clear understanding of Kaizen's value among both employees and management, its actual implementation remains inadequate. Therefore, it can be concluded that Kaizen has a significant impact on organizational effectiveness and employee performance, but its utility is only realized when implemented correctly. This highlights the importance of not only formulating policies and strategies but also fostering active involvement, team spirit, and strong commitment from all employees. Furthermore, close monitoring and evaluation of the Kaizen process at every level of management are crucial to fully realizing its potential benefits.

## 6. Implications

Implications describe how the results contribute to existing knowledge, theory, practice, or policy. Based on these findings, the following sections highlight the policy, practice, and research implications of the current study.

### 6.1. Policy implications

• The findings of this study highlight the importance of supportive policies for Kaizen success in AMIMTDE. The enterprise should establish clear policies to not only advocate for Kaizen but also provide the necessary framework and support for

its successful implementation across its work units. These may include targeted training programs, resource allocation, and leadership development initiatives to ensure that both employees and managers have the competencies required to implement Kaizen effectively.

- It is worth noting that sustaining and enhancing improvements is challenging without incentivizing achievements. That is because designing policies that reward and incentivize active participation in Kaizen activities would boost employee engagement and reinforce the practice of Kaizen. Encouraging consistent participation and ownership of Kaizen initiatives in AMIMTDE requires introducing incentive mechanisms. These could include rewards for employees who make significant contributions to improvement initiatives or demonstrate exceptional commitment to continuous improvement.

- The findings suggest the need to setting standardized metrics for continuously monitoring and evaluating Kaizen implementation. This includes developing formal policies to measure the success of Kaizen practices across different levels of the enterprise, with clear performance indicators to track progress in areas such as productivity, employee engagement, and profitability. Regular reviews and audits of Kaizen practices are essential to ensure they are effectively executed and yielding the expected results. These reviews will help identify areas where implementation is lacking, assess progress, address gaps, and incorporate feedback to make iterative improvements.

## 6.2. Practice implications

- The findings imply bridging the gap between Kaizen's theoretical appeal and practical execution requires empowering employees and aligning Kaizen with the enterprise's broader goals, such as increasing productivity, improving profitability, and enhancing customer satisfaction. By demonstrating how Kaizen contributes directly to these goals, AMIMTDE can foster greater buy-in from both employees and management. Long-term commitment is essential, as Kaizen's success depends on continuous and incremental improvements. Setting long-term goals for Kaizen ensures it remains a sustainable or ongoing practice.

- Effective Kaizen implementation hinges on active or visible leadership. Committed leadership is crucial for success, and managers should engage with employees, motivate them, and provide opportunities for them to contribute ideas and take ownership of improvements.

- The positive impact of Kaizen on employee job performance and affective commitment suggests that actively involving employees in the process – by soliciting their input and ideas – can further increase their sense of ownership and adherence to Kaizen principles.

- The study emphasizes the need to foster sense of ownership, involvement, and engagement among employees. Encouraging participation and promoting teamwork can help bridge the gap between Kaizen's optimistic perception and its actual performance in AMIMTDE.

- The findings also stress the importance of training and development, as continuous learning ensures employees understand Kaizen tools (e.g., 5S) and feel confident in applying them. AMIMTDE should offer Kaizen certification or other professional development opportunities to keep employees updated with best practices and ensure effective Kaizen implementation.

- Given Kaizen's reliance on communication, improving both horizontal (peer-to-peer) and vertical (employee-to-management) communication in the entire enterprise is essential for its successful implementation. Enhancing communication channels will significantly contribute achieve Kaizen goals by maintaining momentum through regular updates on progress.

### 6.3. Research implications

- The findings highlight the need for future research to identify specific barriers to effective Kaizen implementation, such as organizational culture, employee resistance, or lack of management support. It would also be valuable to explore whether the failure to fully implement Kaizen is related to factors like organizational culture, organizational size, or employee demographics.

- Researchers could investigate the long-term effects of Kaizen on organizational effectiveness and employee performance, as short-term results may not fully capture its potential benefits. Tracking performance over time would provide insights into how Kaizen influences organizational culture and employee satisfaction in the long run.

- The findings also underscore the significance of conducting comparative studies that compare organizations that successfully implement Kaizen with AMIMTDE that struggles to offer valuable insights into what practices and factors contribute most significantly to Kaizen's success. Cross-cultural studies would also be useful, as Kaizen's effectiveness may vary across different cultural, organizational, or geographical contexts.

## 7. Recommendations

Successful Kaizen in AMIMTDE requires a holistic approach: committed leadership, empowered employees, continuous learning, and transparent communication. Addressing these pillars while anticipating challenges ensures a sustainable culture of improvement. The following are possible recommendations, typically aimed at practitioners and policymakers proposing ways to apply the research results in real-world settings.

- AMIMTDE should offer regular Kaizen-focused training and development programs at all levels to improve employees' understanding and implementation of Kaizen practices. Encouraging a culture where Kaizen is seen as an ongoing process rather than a one-time effort is crucial for long-term success.

- Piloting Kaizen in specific teams or projects and using data-driven insights will enable a tailored and iterative approach to improving Kaizen effectiveness.

- To increase employee affective commitment, AMIMTDE should acknowledge the contributions employees make to continuous improvement efforts. Publicly recognizing and rewarding individual and team achievements in Kaizen activities boosts emotional investment and morale. Involving employees in decision-making and empowering them to suggest and implement improvements will further increase their commitment by fostering a deeper connection to the enterprise's success.

- To enhance employee job performance, managers should implement clear performance metrics aligned with Kaizen goals and provide regular feedback. It will also help employees understand how their efforts contribute to the enterprise's overall performance and improvement.

- Enhancing team cohesion and effectiveness is essential. AMIMTDE should cultivate a culture of collaboration between employees and management, as well as among employees themselves to facilitate the free flow of ideas for improvement. Cross-functional teams focused on Kaizen initiatives are vital. Regular updates on Kaizen progress, recognition of employees' contributions, and opportunities for refinement will enhance collaborative engagement. A conducive work environment that streamlines communication, reduces barriers, and fosters collaboration ensures employees have the resources and support needed to perform at their best.

- Institutionalizing Kaizen as a core part of AMIMTDE's culture – integrating it into daily operations and long-term strategic goals – will maximize its positive impact on the enterprise's effectiveness.

- To fully realize Kaizen's potential, management must regularly evaluate and adjust its impact on employee performance, affective commitment, and enterprise effectiveness. Strategies should be refined based on feedback and results.

- Investing in essential training programs for employees at all levels can also increase both performance and organizational commitment. Practical steps such as workshops, feedback loops to show tangible outcomes of employee contributions (suggestions and efforts), and collaborative problem-solving sessions where employees at all levels can suggest improvements are essential.

- Managers at all levels must exemplify Kaizen principles and actively foster a culture of continuous improvement, as leadership buy-in is critical to Kaizen's success. They should model and encourage teams to engage in small, incremental improvements, fostering a more proactive and consistent approach to Kaizen. To that effect, they should receive specialized training to model Kaizen values and foster trust and collaboration.

- Managers at all levels should closely and regularly monitor and evaluate Kaizen implementation to ensure continuous adjustment. Regular assessments of Kaizen practices in different departments, identifying barriers, and tracking results will help refine and scale the approach across the entire enterprise.

## Author contributions

**Conceptualization:** Melaku Mengistu Gebremeskel.

**Data curation:** Melaku Mengistu Gebremeskel.

**Formal analysis:** Melaku Mengistu Gebremeskel.

**Funding acquisition:** Melaku Mengistu Gebremeskel.

**Investigation:** Melaku Mengistu Gebremeskel.

**Methodology:** Melaku Mengistu Gebremeskel.

**Project administration:** Melaku Mengistu Gebremeskel.

**Resources:** Melaku Mengistu Gebremeskel.

**Software:** Melaku Mengistu Gebremeskel.

**Supervision:** Melaku Mengistu Gebremeskel.

**Validation:** Melaku Mengistu Gebremeskel.

**Visualization:** Melaku Mengistu Gebremeskel.

**Writing – original draft:** Melaku Mengistu Gebremeskel.

**Writing – review & editing:** Melaku Mengistu Gebremeskel.

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
