## [Decision Letter · Decision Letter 0]

7 Feb 2025

PONE-D-24-57548Full Title: The Role of Kaizen in the Effectiveness of Amhara Metal Industry and Machine Technology Development Enterprise in Ethiopia, as Perceived by the StaffPLOS ONE

Dear Dr. Gebremeskel,

Thank you for submitting your manuscript to PLOS ONE. After careful consideration, we feel that it has merit but does not fully meet PLOS ONE’s publication criteria as it currently stands. Therefore, we invite you to submit a revised version of the manuscript that addresses the points raised during the review process.

We look forward to receiving your revised manuscript.

Kind regards,

Daniel Ebakoleaneh Ufua

Academic Editor

PLOS ONE

Journal Requirements:

Please review your reference list to ensure that it is complete and correct. If you have cited papers that have been retracted, please include the rationale for doing so in the manuscript text, or remove these references 3. and replace them with relevant current references. Any changes to the reference list should be mentioned in the rebuttal letter that accompanies your revised manuscript. If you need to cite a retracted article, indicate the article’s retracted status in the References list and also include a citation and full reference for the retraction notice.

Additional Editor Comments:

based on the comments from the Reviewers, the Author/s are required to make revisions on the manuscript for further consideration

Reviewers' comments:

Reviewer's Responses to Questions

**Comments to the Author**

1. Is the manuscript technically sound, and do the data support the conclusions?

Reviewer #1: No

Reviewer #2: Yes

Reviewer #3: Yes

2. Has the statistical analysis been performed appropriately and rigorously? 

Reviewer #1: I Don't Know

Reviewer #2: Yes

Reviewer #3: Yes

3. Have the authors made all data underlying the findings in their manuscript fully available?

Reviewer #1: Yes

Reviewer #2: Yes

Reviewer #3: Yes

4. Is the manuscript presented in an intelligible fashion and written in standard English?

Reviewer #1: Yes

Reviewer #2: Yes

Reviewer #3: Yes

5. Review Comments to the Author

Reviewer #1: I don't feel like this is completely a research project (no clear control group or pre-/post-comparison) or that it aligns with the science generally published in PLOS One ("natural sciences, medical research, engineering, as well as the related social sciences and humanities"). It seems more consistent with quality improvement and perhaps occupational science or operations management. Its scope is also quite narrow -- a state metal industry and machine technology development enterprise in Ethiopia. Given the topical misalignment and very specific scope, I think it would be better suited for a different journal, and I will defer line edits. I would generally advise line numbers to facilitate future reviews as well as clearer methodologies (e.g, what are sorting, setting, shining, etc.) and making sure that references align with journal preferences. I am not familiar with the statistical methods presented nor are they explained in a manner that I can understand; I would suggest attention here and clarifying Figure 2 as well.

Reviewer #2: Structure and Style: The manuscript is generally well-structured, but some sections, particularly the introduction and discussion, are verbose. Also, the writing is clear but occasionally uses technical jargon that could be simplified for a broader audience. The author should reduce verbosity and ensure consistent use of terminology.

Overall Assessment: The manuscript is a significant contribution to understanding the role of Kaizen in organizational effectiveness within the Ethiopian context. However, several areas, particularly ethics compliance, practical recommendations, and discussion of findings, require substantial revision.

Reviewer #3: The topic is contemporary and important to achieving maximal use of an organization's resources and competitiveness.

The manuscript is quite explanatory and well articulated. The rigorous methodological process is explicit and convincing, especially with the sequential and meticulous process of the analysis. This resonates with the replicability of data analysis in research.

However, the author might need to update the citations and references with more recent literature to capture the current status of Kaizen theory, the ongoing debate, and perhaps the shift in the contemporary business world.

ii. The author needs to state the probable challenges identified in the organization, which could result in resistance to change, inadequate resources for implementation, and the lack of top management support. Identifying these factors will further strengthen the discussion and recommendations for future outlook.

iii. The discussion needs to be more indept to reveal the implications of the findings in relation to recent literature in a manner that the organization will capture what are the wrongs and the beauty of doing the right actions.

iv. Under Ethical Symmentary, the author may avoid the use of pronoun in "the Data were gathered through

questionnaires and document reviews in November and December 2023 personally by me." He could use 'the researcher' instead.

v. Theoretically, the author may need to project why it should be kaizen and not other lean theories. What are the limitations of the theory, and how does it affect the study?

Overall, the manuscript is good and could be considered accepted subject to the minor corrections.

6. PLOS authors have the option to publish the peer review history of their article (what does this mean? ). If published, this will include your full peer review and any attached files.

**Do you want your identity to be public for this peer review?** For information about this choice, including consent withdrawal, please see our Privacy Policy .

Reviewer #1: No

Reviewer #2: **Yes: ** DR. FABUNMI, ABISOLA OLUTOLA

Reviewer #3: No

---

## [Editor Report · Decision Letter 1]

27 Mar 2025

Full Title: The Role of Kaizen in the Effectiveness of Amhara Metal Industry and Machine Technology Development Enterprise in Ethiopia, as Perceived by the Staff

PONE-D-24-57548R1

Dear Dr. Gebremeskel,

We’re pleased to inform you that your manuscript has been judged scientifically suitable for publication and will be formally accepted for publication once it meets all outstanding technical requirements.

Kind regards,

Daniel Ebakoleaneh Ufua

Academic Editor

PLOS ONE
---

## [Editor Report · Acceptance letter]

PONE-D-24-57548R1

PLOS ONE

Dear Dr. Gebremeskel,

I'm pleased to inform you that your manuscript has been deemed suitable for publication in PLOS ONE. Congratulations! Your manuscript is now being handed over to our production team.

Kind regards,

on behalf of

Dr. Daniel Ebakoleaneh Ufua

Academic Editor

PLOS ONE